# Unraveling Depressive Symptomatology and Risk Factors in a Changing World

**DOI:** 10.3390/ijerph20166575

**Published:** 2023-08-13

**Authors:** Rute Dinis Sousa, Ana Rita Henriques, José Caldas de Almeida, Helena Canhão, Ana Maria Rodrigues

**Affiliations:** CHRC, NOVA Medical School, Faculdade de Ciências Médicas, Universidade NOVA de Lisboa, 1169-056 Lisboa, Portugal; anarita.henriques@nms.unl.pt (A.R.H.); caldasjm@nms.unl.pt (J.C.d.A.); helena.canhao@nms.unl.pt (H.C.); ana.m.rodrigues@nms.unl.pt (A.M.R.)

**Keywords:** depression, epidemiology, chronic diseases, multimorbidity, lifestyle factors, COVID-19

## Abstract

Background: This study aimed to examine the prevalence and factors associated with symptoms of depression during the third wave of the COVID-19 pandemic. Methods: A representative sample of Portuguese adults was included in this populational survey, conducted between 25 March and 31 July 2021, with participants completing a structured questionnaire via phone interview. The symptoms of depression were measured using the Portuguese version of the Hospital Anxiety and Depression Scale (HADS). Multivariate logistic regression analyses were used to examine the association between sociodemographic, health, and lifestyle factors and depression levels (normal, mild, or moderate/severe). Results: The estimated prevalence of depression symptoms among participants was 24%. Participants who were women, were in older age groups, had multimorbidity, lived in isolated Portuguese regions such as islands and Alentejo, and were retired or unemployed more frequently reported depression symptoms. Economic hardship was also found to be associated with an increased frequency of mild or moderate-to-severe depression. In contrast, higher levels of education, regular alcohol intake, and regular exercise were associated with a lower frequency of depression symptoms. Conclusions: These findings highlight that during the third wave of the COVID-19 pandemic, a high proportion of Portuguese adults reported depression symptoms, particularly the COVID-19-vulnerable strata such seniors, patients with multimorbidity, and people in economic hardship. On the other hand, citizens who performed regular physical exercise reported lower depressive symptomology. Our work contributes to improving the planning of mental health promotion after the COVID-19 pandemic and future emergencies.

## 1. Introduction

The World Health Organization estimates that depression affects more than 264 million people globally, making it the leading cause of disability worldwide. The burden of depression is particularly high in Europe, where it is estimated to affect over 84 million people [1].

The prevalence of depression varies among European countries, with some countries reporting higher rates than others. Studies have shown that the prevalence of depression is generally higher in Northern and Western European countries, such as Sweden, Denmark, and the Netherlands, compared to Southern and Eastern European countries, such as Italy, Greece, and Romania [2,3,4]. 

Factors that may contribute to the variation in depression rates among European countries include differences in access to mental health care, social and cultural factors, and economic and environmental conditions. A study published in the *Journal of Affective Disorders* in 2016 found that the availability of mental health care and the level of stigma associated with mental health disorders may contribute to the variation in depression rates among European countries [5]. In terms of treatment, there are a range of effective interventions for depression. However, access to these interventions can vary significantly among European countries, with some countries having more robust mental health care systems than others [5].

Despite the significant impact of depression on individuals and societies, it is often overlooked and stigmatized [1,5,6]. 

Depression can affect people of all ages and can occur at any point in a person’s lifespan. However, the prevalence and characteristics of depression can vary between different stages of life [7,8].

In adulthood, the prevalence of depression tends to increase, with women being more likely to experience depression than men. Many adults with depression have a history of recurrent episodes, and the disorder is often accompanied by other mental health conditions, such as anxiety disorders and substance use disorders [9,10].

In older age, the prevalence of depression tends to decrease, although it is still a significant concern, particularly in the context of other physical and mental health conditions that are more common in older age. Depression in older adults may be accompanied by cognitive and physical symptoms, such as memory problems and fatigue, and may be more difficult to diagnose and treat due to overlapping symptoms with other conditions [11,12].

The COVID-19 pandemic has had a significant impact on mental health, including depression, worldwide [13]. The stress and uncertainty caused by the pandemic, along with the social and economic disruptions it has caused, have contributed to an increase in mental health conditions, including depression [14]. Several studies have reported an increased prevalence of depression during the first lockdown, particularly among groups that have been disproportionately impacted by the virus, such as healthcare workers and people with pre-existing mental health conditions. However, few studies have estimated the prevalence of depression symptoms in later stages of the COVID-19 pandemic. During this phase, many citizens were tired of the repeated lockdowns and social restrictions, which could have contributed to the increase in stress and depression symptoms. On the other hand, vaccines where staring to be distributed to the vulnerable population strata. 

The present study aimed to estimate the prevalence of depression symptoms during the third wave of the COVID-19 pandemic in Portugal. We also aimed to identify sociodemographic, lifestyle, and health characteristics associated with depressive symptoms.

## 2. Materials and Methods

### 2.1. Study Design 

The cross-sectional data presented in this study were collected from the fourth follow-up evaluation wave of the Epidemiology of Chronic Diseases Cohort Study (EpiDoC 4), between 25 March and 31 July 2021. Since the aim of the study is to estimate the prevalence of depressive symptoms in this specific period, cross-sectional data were analyzed. 

The Epidemiology of Chronic Diseases (EpiDoC) cohort, represented in Figure 1, is composed of randomly selected Portuguese adults (≥18 years old) living in private households, as described elsewhere [15]. Baseline evaluation (EpiDoC 1, 2011–2013) was performed in two phases. The first involved a face-to-face interview performed by trained interviewers with a structured questionnaire on socioeconomic status, chronic noncommunicable diseases, quality of life, and healthcare resource consumption and a screening for rheumatic and musculoskeletal diseases. The second was a clinical appointment performed for all participants who screened positive for rheumatic and musculoskeletal diseases (RMDs), and 20% of people screened were negative for RMDs. The clinical appointments consisted of a structured evaluation, laboratory tests, and imaging exams, if needed, to establish an RMD diagnosis and evaluate disease-related information [15]. For the follow-up waves EpiDoC 2 (2013–2015), EpiDoC 3 (2015–2016) and EpiDoC 4 (2021), data were collected using a structured questionnaire conducted through phone call interviews using a computer-assisted personal interview system.

#### 2.1.1. Outcome Definitions and Measurements

Symptoms suggestive of depression were measured using the Portuguese version of HADS—Hospital Anxiety and Depression scale—assessed via a phone call interview.

HADS consists of two subscales, one measuring anxiety, with seven items, and one measuring depression, with seven items, with each scored separately. Each item was answered by the patient on a 4-point (0–3) response category scale, so the possible scores ranged from 0 to 21 for anxiety and 0 to 21 for depression. The HADS manual indicates that a score between 0 and 7 is “normal”, between 8 and 10 is “mild”, between 11 and 14 is “moderate”, and between 15 and 21 is “severe” [16]. As we did not have many participants with severe depression for analysis, the moderate and severe levels were grouped together.

#### 2.1.2. Covariates of Interest

We considered the sociodemographic variables sex, age groups, and geographic location, according to NUTS II territorial units (Lisboa e Vale do Tejo, Norte, Centre, Algarve, Alentejo, Madeira and Azores). Age groups were defined as follows: 25–34, 35–44, 45–54, 55–64, 64–65, 65–74 and ≥75 years old. Marital status was categorized as married (married or consensual union) and other (single, widowed, or divorced). Education level was categorized according to the years of education completed: <4 years (less than primary education), 5–9 years (primary or secondary education), 10–12 years (secondary education), and ≥13 years (higher education). Employment status was categorized as follows: active worker (full/part-time), retired, and other (student, unemployed, temporary work disability, and unpaid household worker).

Body mass index (BMI) was categorized as underweight or normal (≤24.99 kg/m^2^), overweight (≥25 and ≤29.99 kg/m^2^), and obese (≥30 kg/m^2^), according to self-reported height and weight. Lifestyle variables such as alcohol intake (never, occasionally, and daily intake) and smoking habits (never, in the past, occasionally, or daily) and regular exercise/sports (yes, no, or occasionally) were noted.

Multimorbidity was defined as having ≥2 self-reported chronic non-communicable diseases of the following list of diseases that were considered: high blood pressure, high cholesterol, cardiac disease, diabetes mellitus, chronic lung disease, problems in the digestive tract, neurological disease, cancer, and rheumatic disease. 

COVID-19 infection, (Have you had COVID-19?—yes, no, do not know/no answer).

#### 2.1.3. Statistical Analysis

Sociodemographic data, lifestyles, and self-reported non-communicable diseases were characterized for the study population. Categorical variables were described as absolute frequency and percentage.

To assess the association between sociodemographic (gender, age group, NUTS II, marital status, education level, and employment status), health (multimorbidity and COVID-19), and lifestyle (alcohol consumption, smoking habits, and exercise) factors with the different depression levels, we performed multinomial logistic regression analysis in two steps. First, we conducted a univariate analysis to select the variables associated with the different depression levels (Appendix A), while considering a significance level of 0.25 to avoid the early exclusion of potentially important variables. Then, through a backward conditional method, we sequentially excluded non-statistically significant variables and compared the models though likelihood ratio tests. The Relative Risk Ratio (RRR) was estimated for each variable with a 95% confidence interval (CI).

All analyses were performed using STATA V17 considering a level of significance of 0.05.

#### 2.1.4. Research Questions 

−What is the prevalence of depressive symptoms in the Portuguese adult population after the lockdown resulting from the COVID-19 pandemic?−What are the sociodemographic factors, lifestyles, and health characteristics associated with depressive symptoms in the Portuguese adult population after the lockdown resulting from the COVID-19 pandemic? −How are these associated with different levels of depressive symptoms? 

We hypothesize that during the third wave of the COVID-19 pandemic, the prevalence of depression symptoms was high and depression symptoms were differently distributed among citizens according to their sociodemographic, lifestyle, and health characteristics.

## 3. Results

### 3.1. Symptoms of Depression According to Sociodemographic Factors, Lifestyle, and Health Characteristics

In our community-based sample, the prevalence of depression symptoms is 12%. Symptoms suggestive of depression are more frequent in women than in men at all depression levels (Figure 2).

There is an increased frequency of depressive symptoms with age, especially in the age group 75+, where 10% of the participants have mild symptoms (Figure 3).

Regarding geographical distribution, the highest levels of depression are found on the islands and in Alentejo (Figure 4). 

Participants with a low education, financial difficulties, and more fragile professional situations (unemployed, temporary work disability) have higher moderate/severe depression levels. Depression is also more common in participants with multiple chronic and non-communicable diseases (multimorbidity) (Table 1).

### 3.2. Factors Associated with Different Levels of Depression

Table 2 presents the final multinomial logistic regression model for the factors associated with depression levels.

A “Normal” depression level was used as reference class. Women (RRR: 1.85 95% CI: 1.26–2.71); people aged 35–44 years (RRR: 3.02 95% CI: 1.13–8.06); people in certain geographic areas (Porto (RRR: 1.68 95% CI: 1.06–2.67), Centro (RRR: 1.84 95% CI: 1.14–2.97), Algarve (RRR: 2.38 95% CI: 1.06–5.33), and Azores (RRR: 1.78 95% CI: 1.00–3.15)); people of certain employment status (retired (RRR: 1.87 95% CI: 1.10–3.16) or other (student, unemployed, temporary work disability, or unpaid household worker)) (RRR: 2.33 95% CI: 1.52–3.58); and people in economic hardship (living with the present income (RRR: 1.58 95% CI: 1.03–2.43), finding it difficult to live with the present income (RRR: 2.48 95% CI: 1.51–4.08), and finding it very difficult to live with the present income (RRR: 4.64 95% CI: 2.58–8.37 and multimorbidity)) (RRR: 2.27 95% CI: 1.54–3.35) showed a significantly higher risk of having symptoms suggestive of mild depression. On the other hand, alcohol intake (daily (RRR: 0.55 95% CI: 0.35–0.85) or occasional (RRR: 0.66 95% CI: 0.47–0.95)) and exercise (regular (RRR: 0.41 95% CI: 0.27–0.62) or occasionally (RRR: 0.67 95% CI: 0.45–0.98)) showed a significantly lower risk of having symptoms suggestive of mild depression.

Women (RRR: 2.16 95% CI: 1.33–3.50); people of certain employment status (retired (RRR: 3.54 95% CI: 1.90–6.58) or other (student, unemployed, temporary work disability, and unpaid household worker)) (RRR: 3.84 95% CI: 2.25–6.55); and people in economic hardship (finding it difficult to live with the present income (RRR: 3.12 95% CI: 1.68–5.79) and finding it very difficult to live with the present income (RRR: 6.84 95% CI: 3.41–13.72)) showed a significantly higher risk of having symptoms suggestive of moderate/severe depression. People with a higher education level (>12 years) (RRR: 0.19 95% CI: 0.07–0.51); who have higher alcohol intake (daily (RRR: 0.43 95% CI: 0.25–0.75) or occasional (RRR: 0.56 95% CI: 0.36–0.87)); and exercise (regular (RRR: 0.44 95% CI: 0.27–0.71) or occasionally (RRR: 0.56 95% CI: 0.34–0.93)) showed a significantly lower risk of having symptoms suggestive of moderate/severe depression.

## 4. Discussion

Our study sheds light on three groups of important findings that will be discussed below: (1)There was a high prevalence of depressive symptoms (12%) in the late phase of the pandemic.(2)There are population groups that are particularly vulnerable to depressive symptoms: the elderly (who remain more vulnerable to COVID-19 even after vaccination) and people living in isolation.(3)There is potentially an important role of physical exercise as a protective agent.

Our results show that during the third wave of the COVID-19 pandemic, 12% of the participants presented mild-to-severe depressive symptoms. This finding is in line with an international study, comprising 78 countries, on the impact of COVID-19 lockdowns on mental health [17], and also one of the latest systematic reviews on mental health in Europe during the COVID-19 pandemic [18].

In Portugal, previous studies estimate that between 6% and 10% of the population experience depression at some point in their lives [19,20,21].

One example is a study published in the *Journal of Affective Disorders* in 2015, which found that the prevalence of depression in Portugal was 7.7%. This study used a representative sample of the Portuguese population and found that the prevalence of depression was higher among women, older individuals, and those with lower levels of education [19]. Another study published in the *Journal of Affective Disorders* in 2018, which investigated the prevalence of depression in Portugal among older adults, found that the prevalence of depression was 9.4% [20].

Our study also displays that having had a COVID-19 infection is not significantly associated with having or not having depression symptoms, nor is it associated with different levels of depressive symptoms. Although, on the one hand, this result is not surprising, on the other hand, the literature has placed emphasis on COVID-19 infection regarding mental health conditions [22,23].

Reflecting on our results, they seem to show that the COVID-19 pandemic had an impact on the mental health of individuals, not because of the reasons for the infection, but because of the social context that it brought about, with the forced and prolonged confinement and the social and economic consequences of it. Daly and Robinson in February 2022 [24] stated that symptoms declined significantly over time and were indistinguishable from pre-pandemic symptom profiles within a few months of the outbreak.

Our results put forward that several factors are associated with different levels of depression among the Portuguese population. In general, this study found that women, older age groups, people from certain geographic regions, and those who were retired or unemployed were at an increased risk of experiencing mild or moderate-to-severe depression. In contrast, higher levels of education, regular alcohol intake, and regular exercise were associated with a lower risk of experiencing mild or moderate-to-severe depressive symptoms.

Our study also found that economic hardship, as measured by difficulty in making ends meet, was associated with an increased risk of experiencing mild or moderate-to-severe depression.

In short, factors such as gender, age, and sociodemographic and economic characteristics were found to have an impact on levels of depression symptoms and are in line with the international literature.

Regarding sex, depression was found to be more frequent in women, which is in line with several international studies [25,26,27,28]. The main conclusion of the studies on depression and gender is that women are more likely to experience depression than men. This finding is consistent across multiple studies and has been observed in different countries and cultures. The reasons for this gender difference in depression are complex and not fully understood, but may involve a combination of biological, psychological, and sociocultural factors. Possible explanations include differences in brain chemistry, hormone levels, coping styles, life stressors, and social roles and expectations. Overall, the gender difference in depression underscores the need for gender-sensitive approaches to the prevention, diagnosis, and treatment of depression.

Regarding age, our findings also show that there is an increase in symptoms suggestive of depression with age. Other studies around the world show the same. A study conducted in the United States found that the prevalence of depression in adults was higher among women than men, with rates of depression increasing with age [9]. Another study conducted in China found that a history of recurrent episodes of depression, and co-occurring mental health conditions, such as anxiety disorders and substance use disorders, were common among adults with depression [10]. In older age, Blazer et al. conducted a study in the United States and found that the prevalence of depression tended to decrease with age, although it remained a significant concern, particularly in the context of other physical and mental health conditions that are more common in older age [11]. Another study conducted in China found that depression in older adults was often accompanied by cognitive and physical symptoms, such as memory problems and fatigue, and may be more difficult to diagnose and treat due to overlapping symptoms with other conditions [12].

This study also identified several health factors that were associated with an increased risk of depression. For example, participants who reported having multiple chronic noncommunicable diseases (multimorbidity) were more likely to have higher levels of depression, as were those who reported a history of COVID-19 infection. Lifestyle factors, such as daily alcohol intake and a lack of regular exercise, were also associated with higher levels of depression.

It is worth noting that this study used a cross-sectional design, which limits our ability to draw conclusions about causality. It is possible that the factors identified as being associated with depression in this study may be influenced by depression, rather than being the cause of it. Further research using longitudinal designs and more detailed data on potential confounders would be needed to understand the relationships more fully between these factors and depression.

In short, the COVID-19 pandemic may have affected mental health via social and economic disruptions. The pandemic has caused significant social and economic disruptions, including job losses, financial strain, and changes to education and other daily activities. Also, it has led to stress and uncertainty. The COVID-19 pandemic has caused significant stress and uncertainty for many people, including concerns about contracting the virus, financial insecurity, and disrupted routines and social support networks. It has also had a disproportionate impact, since some groups have been disproportionately impacted by the pandemic, including healthcare workers, essential workers, and people with pre-existing mental health conditions. These groups may be at increased risk of developing depression and other mental health problems because of the added stress and challenges they have faced. Other interesting topics to discuss in future work are the mental health of people whose sexual lives as couples were altered during the pandemic [29], as well as the mental health of parents whose children were hospitalized due to COVID-19 [30].

There are several strengths and limitations to this study. Some strengths of the study include using a large, representative sample of the Portuguese population, which increases the generalizability of the findings. The study used a validated tool (the HADS scale) to measure symptoms of depression, which is widely used in research and clinical practice. The study adjusted for a range of potential confounders, including sociodemographic, health, and lifestyle factors, which helps to control for their potential influence on the results.

There are also several potential limitations to this study. The study used a cross-sectional design, which limits our ability to draw conclusions about causality. It is possible that the factors identified as being associated with depression in this study may be influenced by depression, rather than being the cause of it. This study relied on self-reported data, which may be subject to bias, such as recall bias or social desirability bias. The study did not collect detailed data on the severity or duration of depression symptoms, which could impact the interpretation of the results. The study did not examine the potential impact of other mental health conditions or life stressors on the risk of depression, which could influence the results.

Overall, the strengths of this study, including its large sample size and the use of validated measures, suggest that the findings are likely to be reliable.

## 5. Conclusions

In conclusion, this study found that the prevalence of symptoms suggestive of depression is high among the Portuguese population, with higher levels observed among women and older age groups. Factors associated with an increased risk of depression included having multiple chronic non-communicable diseases, daily alcohol intake, and a lack of regular exercise. The study also found that women, younger age groups, certain geographic regions, those who were retired or unemployed, economic hardship, and financial stress and insecurity were associated with an increased risk of experiencing mild or moderate-to-severe depression.

Overall, the findings of this study highlight the need to address the high prevalence of depression in the Portuguese population, particularly among groups that are at increased risk. This may involve increasing access to mental health care and support services, implementing policies and interventions to promote mental health and well-being, and addressing the social and economic factors that contribute to mental health problems.

## Figures and Tables

**Figure 1 ijerph-20-06575-f001:**
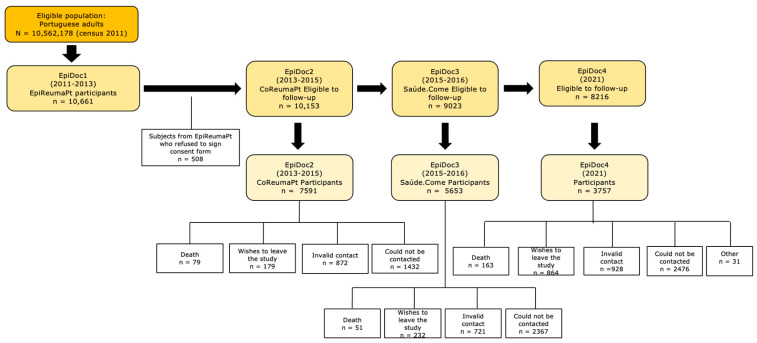
EpiDoC cohort flowchart.

**Figure 2 ijerph-20-06575-f002:**
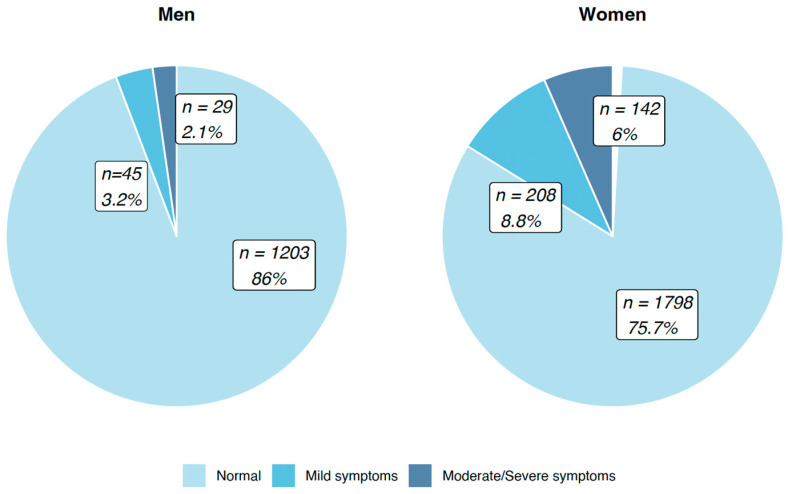
Depression symptom levels according to gender.

**Figure 3 ijerph-20-06575-f003:**
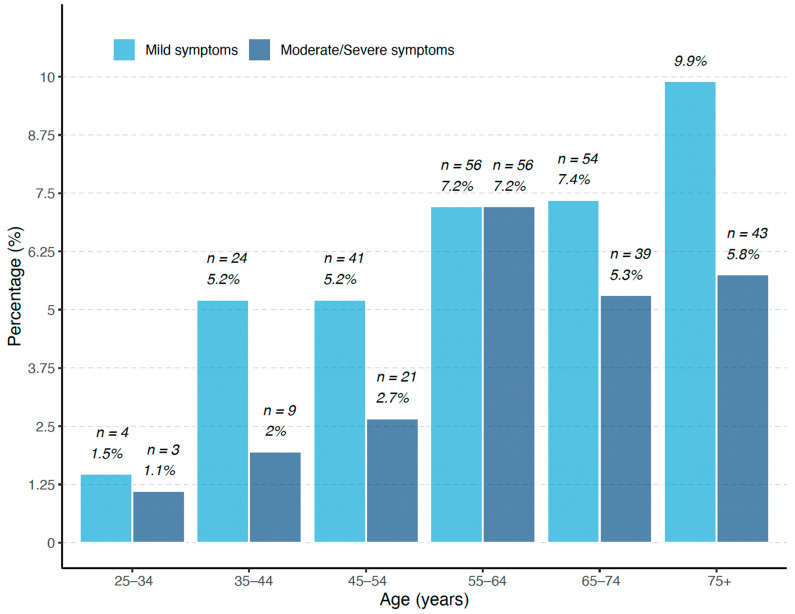
Depression symptom levels according to age group.

**Figure 4 ijerph-20-06575-f004:**
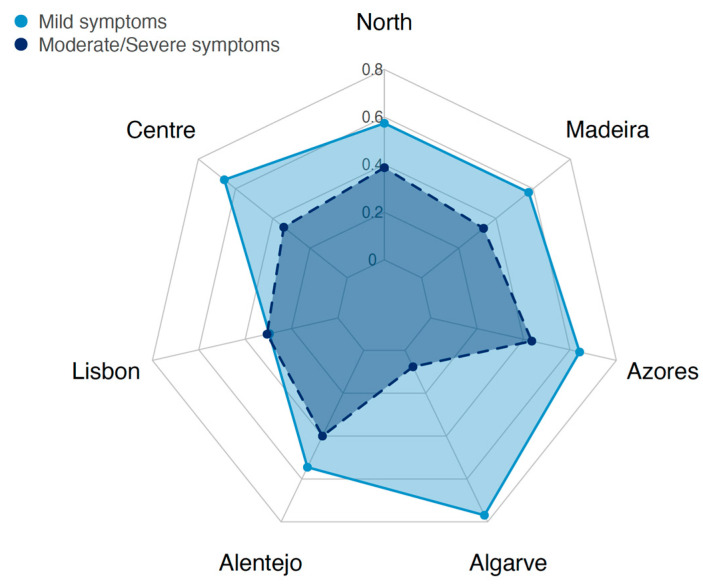
Depression symptom levels according to NUTSII (Portuguese regions).

**Table 1 ijerph-20-06575-t001:** Sociodemographic, lifestyle, and health characteristics in individuals without depression, with mild depression, and with moderate/severe depression.

	All (n = 3425)	Missing (n %)	Normal (n = 3001) (HADS-D 0–7)	Missing (n%)	Mild (n = 253) (HADS-D 8–10)	Missing (n%)	Moderate/Severe (n = 171) (HADS-D 11–21)	Missing (n%)
**Sociodemographic**								
**Marital status**		3 (0.09)		1 (0.03)		2 (0.79)		-
Married	2241 (65.49%)		1969 (87.86%)		160 (7.14%)		112 (5.00%)	
Other	1181 (34.51%)		1031 (87.30%)		91 (7.71%)		59 (5.00%)	
**Education level**		7 (0.20)		5 (0.00)		1 (0.40)		1 (0.58)
0–4 years	1168 (34.17%)		926 (79.28%)		141 (12.07%)		101 (8.65%)	
5–9 years	698 (20.42%)		614 (87.97%)		47 (6.73%)		37 (5.30%)	
10–12 years	746 (21.83%)		682 (91.42%)		38 (5.09%)		26 (3.49%)	
>12 years	806 (23.58%)		774 (96.03%)		26 (3.23%)		6 (0.74%)	
**Employment status**		37 (1.08)		27 (0.90)		4 (0.02)		6 (3.51)
Employed full/part-time	1744 (51.48%)		1650 (94.61%)		67 (3.84%)		27 (1.55%)	
Retired	1158 (34.18%)		966 (83.42%)		111 (9.59%)		81 (6.99%)	
Other	486 (14.34%)		358 (73.66%)		71 (14.61%)		57 (11.73%)	
**Income perception**		57 (1.66)		44 (1.47)		5 (1.98)		8 (4.68)
Living comfortably with the present income	1130 (33.55%)		1076 (95.22%)		37 (3.27%)		17 (1.50%)	
Living with the present income	1597 (47.42%)		1421 (88.98%)		113 (7.08%)		63 (3.94%)	
Finding it difficult to live with the present income	463 (13.75%)		354 (76.46%)		59 (12.74%)		50 (10.80%)	
Finding it very difficult to live with the present income	178 (5.29%)		106 (59.55%)		39 (21.91%)		33 (18.54%)	
**Household size**		46 (1.34)		34 (1.13)		5 (1.98)		7 (4.09)
One person	585 (17.31%)		469 (80.17%)		66 (11.28%)		50 (8.55%)	
Two people	1205 (35.66%)		1046 (86.80%)		93 (7.72%)		66 (5.48%)	
Three or more people	1589 (47.03%)		1452 (91.38%)		89 (5.60%)		48 (3.02%)	
**BMI (kg/m^2^)**		141 (4.12)		106 (3.53)		23 (9.09)		12 (7.02)
Underweight/normal	1237 (37.67%)		1116 (90.22%)		69 (5.58%)		52 (4.20%)	
Overweight	1332 (40.56%)		1180 (88.59%)		89 (6.68%)		63 (4.73%)	
Obese	715 (21.77%)		599 (83.78%)		72 (10.07%)		44 (6.15%)	
**Alcohol intake**		35 (1.02)		25 (0.83)		3 (1.19)		7 (4.37)
Daily	833 (24.57%)		779 (93.52%)		34 (4.08%)		20 (2.40%)	
Occasional	1182 (34.87%)		1091 (92.30%)		58 (4.91%)		33 (2.79%)	
Never	1375 (40.56%)		1106 (80.44%)		158 (11.49%)		111 (8.07%)	
**Smoking habits**		32 (0.93)		23 (0.77)		3 (1.19)		6 (3.51)
Past smoker	758 (22.34%)		698 (92.08%)		36 (4.75%)		24 (3.17%)	
Current/occasional smoker	516 (15.21%)		464 (89.13%)		31 (6.01%)		21 (4.07%)	
Never	2119 (62.45%)		1816 (85.70%)		183 (8.64%)		120 (5.66%)	
**Regular** **physical Exercise**		38 (1.11)		29 (0.97)		4 (1.58)		5 (2.92)
Never	1744 (51.49%)		1451 (83.20%)		177 (10.15%)		116 (6.65%)	
Yes	830 (24.51%)		773 (93.13%)		32 (3.86%)		25 (3.01%)	
Occasionally	813 (24.00%)		748 (92.00%)		40 (4.92%)		25 (3.08%)	
**Health**								
**Multimorbidity (self-reported)**				-		-		-
No	1719 (50.19%)		1615 (93.95%)		62 (3.61%)		42 (2.44%)	
Yes	1706 (49.81%)		1386 (81.24%)		191 (11.20%)		129 (7.56%)	
**COVID-19**								
**COVID-19 infection**		15 (0.44)		15 (0.50)		-		-
No	3145 (92.23%)		2749 (87.41%)		238 (7.57%)		158 (5.02%)	
Yes	265 (7.77%)		237 (89.43%)		15 (5.66%)		11 (4.91%)	

**Table 2 ijerph-20-06575-t002:** Multinomial logistic regression model for the association of sociodemographic, lifestyle, and health characteristics and depression levels.

		Normal vs. Mild	
		Relative Risk Ratio	[95% CI]
Gender			
	Male	Ref	-
	Female	1.85	[1.26–2.71]
Age group			
	25–34	Ref	-
	35–44	3.02	[1.13–8.06]
	45–54	1.49	[0.55–4.03]
	55–64	1.76	[0.64–4.81]
	65–74	1.16	[0.39–3.47]
	≥75 years	1.62	[0.53–4.91]
NUTSII			
	LVT	Ref	-
	Norte	1.68	[1.06–2.67]
	Centro	1.84	[1.14–2.97]
	Alentejo	1.30	[0.64–2.65]
	Algarve	2.38	[1.06–5.33]
	Azores	1.78	[1.00–3.15]
	Madeira	1.45	[0.77–2.73]
Education level			
	0–4 years	Ref	-
	5–9 years	0.77	[0.51–1.17]
	10–12 years	0.96	[0.60–1.53]
	>12 years	0.72	[0.41–1.24]
Employment status			
	Employed full/part-time	Ref	-
	Retired	1.87	[1.10–3.16]
	Other	2.33	[1.52–3.58]
Income perception			
	Living comfortably with the present income	Ref	-
	Living with the present income	1.58	[1.03–2.43]
	Finding it difficult to live with the present income	2.48	[1.51–4.08]
	Finding it very difficult to live with the present income	4.64	[2.58–8.37]
BMI (kg/m^2^)			
	Underweight/normal	Ref	-
	Overweight	1.03	[0.73–1.47]
	Obese	1.16	[0.80–1.70]
Alcohol intake			
	Never	Ref	-
	Daily	0.55	[0.35–0.85]
	Occasional	0.66	[0.47–0.95]
Regular Exercise			
	Never	Ref	-
	Yes	0.41	[0.27–0.62]
	Occasionally	0.67	[0.45–0.98]
Multimorbidity (self-reported)			
	No	Ref	-
	Yes	2.27	[1.54–3.35]
		Normal vs. Moderate/Severe	
		Relative Risk ratio	[95% CI]
Gender			
	Male	Ref	-
	Female	2.16	[1.33–3.50]
Age group			
	25–34 years	Ref	-
	35–44 years	0.86	[0.27–2.79]
	45–54 years	0.75	[0.25–2.21]
	55–64 years	1.63	[0.56–4.70]
	65–74 years	0.58	[0.18–1.89]
	≥75 years	0.66	[0.20–2.20]
NUTSII			
	LVT	Ref	-
	Norte	1.13	[0.69–1.85]
	Centro	0.82	[0.47–1.43]
	Alentejo	0.66	[0.29–1.52]
	Algarve	0.18	[0.02–1.43]
	Azores	0.84	[0.42–1.67]
	Madeira	0.79	[0.38–1.67]
Education level			
	0–4 years	Ref	-
	5–9 years	0.79	[0.50–1.26]
	10–12 years	0.81	[0.46–1.44]
	>12 years	0.19	[0.07–0.51]
Employment status			
	Employed full/part-time	Ref	-
	Retired	3.54	[1.90–6.58]
	Other	3.84	[2.25–6.55]
Income perception			
	Living comfortably with the present income	Ref	-
	Living with the present income	1.38	[0.77–2.47]
	Finding it difficult to live with the present income	3.12	[1.68–5.79]
	Finding it very difficult to live with the present income	6.84	[3.41–13.72]
BMI (kg/m^2^)			
	Underweight/normal	Ref	-
	Overweight	0.84	[0.55–1.27]
	Obese	0.78	[0.49–1.24]
Alcohol intake			
	Never	Ref	-
	Daily	0.43	[0.25–0.75]
	Occasional	0.56	[0.38–0.87]
Regular Exercise			
	Never	Ref	-
	Yes	0.44	[0.27–0.71]
	Occasionally	0.56	[0.34–0.93]
Multimorbidity (self-reported)			
	No	Ref	-
	Yes	1.51	[0.95–2.39]

Normal level as the reference. Statistically significant associations are highlighted in bold.

## Data Availability

The data presented in this study are available on request from the corresponding author. The data are not publicly available due to restrictions that apply to the availability of these data, which were used under a license for the current study.

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
