# Peer review of "Unraveling Depressive Symptomatology and Risk Factors in a Changing World"

_ijerph, 2023, doi:10.3390/ijerph20166575_

Round 1
Reviewer 1 Report
the article proposed by the authors is interesting and investigates some aspects of pandemic covid 19 that have been underestimated.
the authors are requested to enrich the captions of the presented figures and make them homogeneous so that they are clear to the readers.
also it is advisable to better clarify the objectives of the 'article in the abstract and enrich the discussion with this article: Costantini E, Trama F, Villari D, Maruccia S, Li Marzi V, Natale F, Balzarro M, Mancini V, Balsamo R, Marson F, Bevacqua M, Pastore AL, Ammirati E, Gubbiotti M, Filocamo MT, De Rienzo G, Finazzi Agrò E, Spatafora P, Bisegna C, Gemma L, Giammò A, Zucchi A, Brancorsini S, Ruggiero G, Illiano E. The Impact of Lockdown on Couples' Sex Lives. J Clin Med. 2021 Apr 1;10(7):1414. doi: 10.3390/jcm10071414. PMID: 33915856; PMCID: PMC8037775.
moderate English revision
Author Response
Thank you for your comments.
All the captions have been improved and homogenized to make them easier to understand.
The objectives of the 'article in the abstract were better clarified: “The study aimed to examine the prevalence and factors associated with symptoms of depression during the third wave of COVID 19 pandemic in Portugal.”
Costantini E, Trama F, Villari D, Maruccia S, Li Marzi V, Natale F, Balzarro M, Mancini V, Balsamo R, Marson F, Bevacqua M, Pastore AL, Ammirati E, Gubbiotti M, Filocamo MT, De Rienzo G, Finazzi Agrò E, Spatafora P, Bisegna C, Gemma L, Giammò A, Zucchi A, Brancorsini S, Ruggiero G, Illiano E. The Impact of Lockdown on Couples' Sex Lives. J Clin Med. 2021 Apr 1;10(7):1414. doi: 10.3390/jcm10071414. PMID: 33915856; PMCID: PMC8037775.This reference was added in the discussion section.
Reviewer 2 Report
- This is a large cross-sectional study of depressive and anxiety symptoms in the general population in the pandemic.
There is a mistake in the abstract stating that youmnger age was found to be connected to higher level of depressive symptoms. The same in the discussion section. It is not in agreement with the Figure 2. - Research questions are put forward; however, the hypothesis is lacking.
- The manuscript is clear and well-structured.
- More recent bibliography should be used whenever possible.
- The methods are described in-detail. Nonetheless, study design is not enough clearly presented. I would use a study flowchart beginning from baseline evaluation of the population. Is that true that in the fourth wave of the EpiDoC study all RMDs and only 20% of non RMDs subjects were included. Is true, this is nor representative for general population. Please, make it clear for the reader. Moreover, the abbreviation RMDs is not explained.
- The tables are appropriate.
- Conclusions are consistent with the evidence.
- In the discussion section I lack more possible explanations for the results.
Author Response
Thank you for your comments.
There is a mistake in the abstract stating that youmnger age was found to be connected to higher level of depressive symptoms. The same in the discussion section. It is not in agreement with the Figure 2. – this was corrected in both sections, and the word younder was replaced by older.
Research questions are put forward; however, the hypothesis is lacking. – we entered the hypothesis after the research questions: We hypothesise that depressive symptoms are associated with socio-demographic, lifestyles, and health factors.
Study design is not enough clearly presented. I would use a study flowchart beginning from baseline evaluation of the population. A flowchart explaining the complete cohort, including all waves, was included as figure 1.
Is that true that in the fourth wave of the EpiDoC study all RMDs and only 20% of non RMDs subjects were included. Is true, this is nor representative for general population. Please, make it clear for the reader. No. This study involved a face-to-face survey performed by trained interviewers at the household of 10,661 subjects who were randomly selected by a stratified multistage sampling. The second phase of the first wave, participants who screened positive (64%) for at least one RMD as well as 20% of individuals with a negative screening were invited for assessment by a rheumatologist. In total, 3,877 subjects participated in this second phase of the first wave, where they were also invited to donate a blood sample.
This confusion is cleared up with the introduction of the cohort flowchart.
Moreover, the abbreviation RMDs is not explained. – RMD stands for Rheumatic and musculoskeletal diseases. This was added in the paper.
In the discussion section I lack more possible explanations for the results. Thank you, more plausible reasons to support the results have been added, namely differences in gender, age and economic status.
Reviewer 3 Report
Thanks to the authors. An extensive study has been carried out. I recommend that they give information about the number of samples in the summary section. As the authors mentioned, there has been an increase in the susceptibility of many people to depression during the pandemic process. I suggest that the authors also consider this study, in which parents whose children were hospitalized during the pandemic were psychologically affected. Yildiz et al., 2022, https: // doi.org/10.3390 /children9101448
Author Response
Thank you for your comments.
I recommend that they give information about the number of samples in the summary section.
A detailed flowchart describing the entire cohort was introduced in the methods section, so the number of participants is clearer.
I suggest that the authors also consider this study, in which parents whose children were hospitalized during the pandemic were psychologically affected. Yildiz et al., 2022, https: // doi.org/10.3390 /children9101448 This reference was added in the discussion section: Yildiz, E.; Koc Apaydin, Z.; Alay, B.; Dincer, Z.; Cigri, E. COVID-19 History Increases the Anxiety of Mothers with Children in Intensive Care during the Pandemic in Turkey. Children 2022, 9, 1448. https://doi.org/10.3390/children9101448